# Atmospheric Cold Plasma as an Alternative to Chlorination in Soft Wheat Flour to Prepare High-Ratio Cakes

**DOI:** 10.3390/foods13152366

**Published:** 2024-07-26

**Authors:** Shikhadri Mahanta, Jayne Bock, Andrew Mense, Nahndi Kirk-Bradley, Joseph Awika, Janie McClurkin Moore

**Affiliations:** 1Department of Biological and Agricultural Engineering, Texas A&M University, College Station, TX 77843, USA; shikhadri@tamu.edu (S.M.); ntkirkbr@tamu.edu (N.K.-B.); 2Wheat Marketing Center, Portland, OR 97209, USA; jbock@wmcinc.org (J.B.); amense@wmcinc.org (A.M.); 3Department of Food Science and Technology, Texas A&M University, College Station, TX 77843, USA; joseph.awika@ag.tamu.edu

**Keywords:** atmospheric cold plasma, soft wheat flour, chlorination, high-ratio cake

## Abstract

Chlorination is a common chemical modification process of soft wheat flour to prepare high-ratio cakes. Due to safety and labeling concerns of flour chlorination, alternatives to chlorination have been researched. Atmospheric Cold Plasma (ACP) is an emerging technology which is applicable for a wide range of food and biological components, including cereal grain products. The potential of ACP as an alternative to chlorination for high-ratio cakes has not been researched. Soft wheat flour was treated at 50 kV, 60 kV, and 70 kV each for 5, 6, and 7 min and compared to untreated and chlorinated wheat flour. High-ratio cakes were prepared from the chlorinated, treated, and untreated soft wheat flour and their properties were compared. Changes in the flour properties and the high-ratio cakes were observed at different treatment conditions. It was found that after 50 kV, 6 min, 50 kV, 7 min and 60 kV, 6 min had the better flour pasting properties, higher cake volume, and better texture properties as compared to untreated wheat flour and chlorinated wheat flour. This determines the potential of the application of ACP as an alternative to chlorination or to reduce the use of chlorination in soft wheat flour.

## 1. Introduction 

Wheat is a staple cereal or used as the base of several major diets in various regions across the world, commonly in developing countries but also significant in the developed countries [1]. According to FAO reports, total wheat uses in 2023–2024 stand at 794 million tons, which has increased by 15.4 million tons, i.e., 2 percent growth rate since 2022–2023 [2].

Bread, cookies, confectionery items, noodles, and textured wheat gluten, often known as seitan, are all made from wheat [3,4,5,6]. The main benefit of wheat over other temperate crops is due to the viscoelastic properties of gluten when the flour is mixed into a dough along with its starchy endosperm, which enable it to be processed into a variety of breads and other baked goods including cakes and cookies, pasta, and noodles, as well as other processed meals. The viscoelasticity of wheat-based doughs is determined by the arrangements and interactions of the grain storage proteins, which collectively make up the protein fraction known as “gluten” [7].

The terms “soft” and “hard” in reference to wheat are descriptors of the kernel’s texture. Flour produced from hard wheat can be used to produce bread flour, which has stronger gluten strength compared to soft wheat, which is used for cake, or pastry flour. Soft and hard wheats provide flour with extremely distinct and clearly quantifiable qualities. The average particle size of flour made from soft wheat kernels is less than that of flour made from hard wheat kernels. Soft wheat flour often has less protein than hard wheat flour. The low protein content (~7–10% protein, 14% mb) of soft wheat flour is ideal for baking cakes, cookies, pastries, snack foods, and breakfast biscuits. Contrasted to hard wheat, which is used almost exclusively for breads, soft wheat yields a greater quantity and variety of goods. All these products have a better appearance and eating quality when prepared using soft wheat flour as opposed to hard wheat flour. Soft wheat results in greater volume and a more tender texture in these foods than hard wheat. Soft wheat products remain popular globally [8,9,10]. Along with low ash and low protein content, soft wheat flours usually have a low water absorption rate which can be blended into a smooth cake batter and tender cakes [11]. 

Cake is a type of semi-dried foam dessert with air pockets wrapped in a network of proteins, lipids, and carbohydrates. A high-ratio cake (HRC) has a smooth batter as the fat is finely and evenly dispersed through the aqueous phase [12]. A HRC is prepared from defined recipes with fluid mediums that have expanded due to gas created by chemical leavening agents dissolved in the medium. Cake flour has a low protein content, low ash content, and finer particle size. The finer particle size of the flour helps in creating a smooth batter, resulting in a finer crumb structure in the finished cake [13].

Chlorination and heat treatment are two commonly used flour modification processes that can help improve baking outcomes of the HRC. Soft wheat flour has been chlorinated since the early 1930s, which is required to manufacture HRCs with the best possible quality attributes [14]. HRCs prepared with optimally chlorinated flour at optimal absorption levels have an enhanced volume, a finer, more uniform grain, a whiter crumb color, improved symmetry [15], and improved sensory characteristics [16]. However, the precise mechanisms are yet unknown [17]. A quick, surface-dependent reaction occurs when chlorine gas and flour are combined [18]. Chlorine is commercially applied in gaseous form at the rate of 300–1500 mg/kg of flour, and it reacts with the moisture present in the flour [19]. The application of chlorine in wheat flour can be summarized by the formula
(1)$${\mathrm{Cl}}_{2}+3H_{2}O\;\to \;2H_{3}O^{+}+{\mathrm{OCl}}^{-}+{\mathrm{Cl}}^{-}$$

Almost all the ingredients in flour—gluten, starch, lipids, water soluble, and pentosans—are chemically altered throughout the chlorination process [20,21]. Changes in the molecular structure of protein, starch, and other components of wheat flour increase gel hydration ability and starch pasting, and they reduce gluten strength. Worker safety and public health concerns arise from the usage of chlorine gas in a grain mill. Chlorination in flour can lead to the formation of harmful byproducts such as chlorinated hydrocarbons and chlorinated benzenes [19]. The by-products have adverse health effects including potential carcinogenic effects and disruptions to the endocrine system [22]. These concerns underscore the need for alternatives to chlorine to treat soft wheat flour. Alternatives to the chlorination of soft wheat flour, such as high heat treatment, and additional ingredients like xanthan gum and hydrogen peroxide, have also been explored. In EU countries, the chlorination of wheat flour has been replaced to some extent by heat-treated flour, which is used to make HRCs. Protein denaturation and partial starch gelatinization occur during heat treatment of the wheat flour, increasing the viscosity of the batter [23]. Significant improvement in the baking performance of cakes with soft wheat flour has been explored with both dry and hydrothermal heat treatments [24]. However, there are certain disadvantages with the high-heat treatment of wheat flour. It is necessary to optimize the heat processing time and temperature to avoid formation of carcinogenic compounds like acrylamide, a by-product of the Maillard reaction [25]. Lowering the treatment temperature and time alternatively may reduce the desired compounds formed during the Maillard reaction [26]. During microwave heating, the rapid increase in temperature can lead to a weaker gluten structure, increased starch granule rupture, and a reduction in the swelling ability of the wheat flour [27,28]. 

New processing treatments, such atmospheric cold plasma (ACP), could be utilized to modify wheat flour. ACP is a novel technology that can generate changes in the protein and starch structure of wheat flour while producing no hazardous residues [29]. This advanced technology can be applied to different food materials to bring desirable changes in its biochemical profile. Plasma generated during ACP treatment is a state of high energy, generating reactive gas species which can interact at a molecular level [30]. A study previously established that ACP causes protein oxidation, which increases the dough strength of bread wheat flour and modifies its functionality. Another study by, demonstrated changes in the secondary structure of wheat proteins, where an increase in α-helix and a decrease in β-sheets were observed [31,32]. The oxidative potential of ACP, which produces ozone, can be used to reduce the use of oxidative agents such as potassium bromate, chlorine, and benzoyl peroxide, which are commonly utilized in the industry to enhance the functionality of flour [32]. The use of ozone in the grain industry has been extended due to its effectiveness in reducing microbial contamination [33]. It also affects the milling ability of the grains; however, the treatment is intended for the end product, i.e., wheat flour.

The aim of this research was to understand the changes generated in soft wheat flour after treatment with ACP. The baking ability of the flour treated at different rates of ACP was explored and compared with chlorinated flour.

## 2. Materials and Methods

### 2.1. Materials

An untreated soft wheat flour sample was procured from Grain Craft, Pendleton, OR, USA, and all the reagents used were of analytical grade. All treatments were replicated 3 times.

### 2.2. Methods

#### 2.2.1. ACP Treatment of Soft Wheat Flour

Atmospheric Cold Plasma generated through the Dielectric Barrier discharge (DBD) method was used for treating the flour samples. The DBD has two electrodes of six inches in diameter, connected to transformers to generate high-voltage electricity. The anode is known as the high-voltage electrode and both the electrodes are enveloped by a plastic insulator [34]. The rate of voltage generated can be controlled as required in the DBD plasma generation method. A total of 100 g of wheat flour was taken in a 1.5–1.7 mils thick LDPE plastic bag of 26.6 cm × 27.3 cm and filled with modified atmospheric gas with 65.03% oxygen, 5.05% nitrogen, and 29.92% carbon. The modified atmospheric gases were obtained from Airgas, Inc. (Radnor, PA, USA). A height of 3.5–4 cm was maintained in the bags to ensure enough headspace for plasma generation. The samples were placed in the plasma generation chamber and treated at 50 kV, 60 kV, and 70 kV each for 5, 6, and 7 min. Higher rates of ACP treatment were used since more changes in wheat flour properties were observed at a higher rate of ACP treatment [31,32]. The rates of plasma treatment were optimized by running a methylene blue test [35]. After plasma treatment, the treated wheat flour was allowed to be exposed to the reactive gas species in the Ziploc bag for 12 h and later transferred to an airtight container. A schematic of the ACP system to treat wheat flour is shown in Figure 1. 

#### 2.2.2. Analysis of Wheat Flour 

Analyses of the untreated, ACP-treated, and chlorinated wheat flour samples were performed to understand the effect of different treatments. Moisture, protein, and pH, respectively, were measured using Approved Methods 44-15.02, 46-30.01, and 02-52.01 (AACC International 1999) [36]. The samples were further analyzed for color and pasting properties.

#### 2.2.3. Color Measurement of the Soft Wheat Flour

The color of the flour samples was measured by a CR-410 Chroma Meter (Konica Minolta Sensing Americas, Inc., Ramsey, NJ, USA) at room temperature. The data were recorded using a software program (Spectra Magic NX Professional/Lite v2.5). The equipment was calibrated with standard reference tiles. The sample container was uniformly filled with samples, and the color values were measured on an *L* a* b** scale. The samples were measured in triplicates, and the average value of *L**, *a**, and *b** was taken for determining the differences in the different treated samples. The differences in color samples were calculated by Equation (2):(2)$$∆E^{*}=\surd {\left(∆L^{*}\right)}^{2\;}+{\left(∆a^{*}\right)}^{2\;}+{\left(∆b^{*}\right)}^{2\;}$$

Here, $∆E^{*}$ is the difference in color between the samples, indicating the whiteness index of the samples. The values
$$∆L^{*}=L^{*}\left(t\right)-L^{*}\left(0\right)\;$$
$$∆a^{*}=a^{*}\left(t\right)-a^{*}(0)$$
$$∆b^{*}=b^{*}\left(t\right)-b^{*}(0)$$
are used in the calculation, where (*t*) is the color of the treated samples, and (0) is the color of the untreated samples [37].

#### 2.2.4. Pasting Properties Analysis of Soft Wheat Flour

The starch pasting properties of the flour samples were determined by the Rapid Visco Analyzer (RVA) (RVA 4500, Perten Instruments of Australia Pty Ltd., Macquarie Park, NSW, Australia). The method used employs a temperature profile (Table 1) that more closely mimics the internal temperature rise encountered internally by an HRC during baking. Flour (3.5 g, 14% mb) was added to 25 g of 50% *w*/*w* pre-dissolved sucrose solution in the sample canister and thoroughly mixed with the paddle. The pasting viscosities and temperature were calculated with a software program (Thermocline for Windows, TCW v.3.0). The 50% *w*/*w* sucrose solution was used instead of water to mimic the solvent conditions experienced by the flour during cake baking. For each sample, tests were conducted in triplicates.

### 2.3. Baking of High-Ratio Cake

The AACC international method 10-90.01 was used for cake baking [36]. The HRC were baked in triplicate. The water absorption was optimized to 125% absorption with respect to the flour. The optimum absorption was the level that produced the best volume and crumb grain. The amount of baking powder adjusted to the formulation was based on the laboratory elevation and barometric pressure on the day of the bake, corrected to sea level. The formulation for the HRC is shown in the Table 2.

The HRCs were baked at 375 °F for 23 min in a reel oven. The cooled HRCs were removed from the pans and allowed to sit at room temperature for 3 h and later packed in Ziploc bags for analysis the next day.

### 2.4. Analysis of High-Ratio Cake

#### 2.4.1. High-Ratio Cake Volume, Texture, and C-Cell Structure Analysis

The volumes of the HRCs were measured by Vol Scan, which uses laser topography, to model the dimensions of a tested baked product at the Wheat Marketing Center, Portland. The AACC method 10-16.01 was used to determine the volume and dimensional profile of the HRC [38]. The arms of the instrument were adjusted according to the height of the HRCs, and the HRCs were mounted on the longest axis and inserted in the support pins. The HRCs were scanned by laser equipment and at the end of the test, the dimensional profiles of the HRCs were automatically identified by the software and displayed.

Texture was measured using a texture analyzer (TA-XT2, Texture Technologies, Scarsdale, NY, USA). A 1-inch diameter cylinder probe was used to determine the hardness of the baked HRC. Test conditions for the texture profile analysis were 4.0 mm/s pre-test speed; 1.7 mm/s test speed; 4.0 mm/s post-test speed; strain set at 40%; and a trigger force of 5 g. Each HRC was measured for texture at two positions on the cake surface and the average value of the firmness was taken. Texture analysis of the HRC was conducted on the day following the baking.

C-cell analysis of the HRC crumb structure was conducted following the AACC 10-18.01 method. C-cell analysis was available in monochrome and color models to determine the internal structure of baked products. For each sample, a cake slice was prepared by cutting through the diameter of the HRC at its center. A template was used to ensure consistent slice thickness and cell structure analysis. The crumb of the cut slice of the HRC was used in the test [38].

#### 2.4.2. Statistical Analysis

All the data were statistically analyzed in SAS 9.4 software. One-way ANOVA test and Tukey HSD post hoc test were conducted for each parameter to determine statistical significance amongst the different sets of data. The Tukey HSD post hoc test helps to determine exactly which pair of treatments were significantly different from each other.

## 3. Results

### 3.1. Soft Wheat Flour Properties

#### 3.1.1. Moisture and Protein Content and pH of Soft Wheat Flour

The moisture content, protein content, and pH value of the soft wheat flour samples are provided in Table 3 below.

#### 3.1.2. Color of Soft Wheat Flour

The color of the flour samples measured using a colorimeter provides the *L*, *a*, and *b* values. The ∆E value was calculated for different samples to provide a cumulative value for the *L*, *a*, and *b* values in Table 4. ∆E values greater than 0.5 indicate a noticeable difference in the color of the samples [37]. ∆E values were higher than 0.5 in all cases when compared to untreated soft wheat flour samples. The ∆E value was highest in the chlorinated flour sample while it was lowest at 50 kV, 5 min ACP-treated soft wheat flour samples. ∆E values were seen to be increasing with an increase in the rate of ACP treatment.

#### 3.1.3. Pasting Properties of Soft Wheat Flour

A viscosity graph was developed for each sample, and viscosity values at different points were determined, which is shown in Table 5 below. The initial viscosity, Trough 1, Breakdown viscosity, and the peak viscosity of the flour samples were obtained from the RVA. The temperature profile was established to generally replicate the internal temperature rise of the cake batter during baking. The pasting properties of soft wheat flour under different treatment conditions are shown in Table 5.

### 3.2. High-Ratio Cake Properties

#### 3.2.1. High-Ratio Cake Volume

The volume of the HRC measured with Vol Scan provides the cake volumes in milliliters. The volume of the HRCs was lowest for the untreated soft wheat flour. An increase in the volume of the HRCs with an increase in the rate of ACP treatment was observed, which is presented in Figure 2. The highest volume of the HRCs was obtained at 70 kV, 6 min of APC treatment.

#### 3.2.2. High-Ratio Cake Texture

The firmness of the HRC determines its textural properties, which is shown in Figure 3. The firmness of the HRCs was lowest at 60 kV, 7 min of ACP treatment while chlorinated wheat flour had the highest firmness. A decreasing trend in firmness of the HRCs was observed till 60 kV, 7 min of ACP treatment. The firmness of the HRCs increased from 60 kV, 7 min of ACP treatment.

#### 3.2.3. C-Cell Structure Analysis of High-Ratio Cakes

C-cell analysis of the HRC determines the crumb color and cell properties. According to the slice area taken for the evaluation of the number of cells, the area of cells in percentage, number of holes, area of holes in percentage, volume of cells, non-uniformity of the cells, cell diameter, and crumb fineness were determined. From all the available parameters, the average cell diameter was considered for the different treatment conditions to understand uniformity in the crumb structure [39]. The cell diameter in the crumb at different treatment conditions is presented in Table 6.

## 4. Discussion

### 4.1. ACP Treatment of Soft Wheat Flour

The amount of wheat flour taken in the Ziploc bag and the rate of plasma treatment for each treatment were determined through a methylene blue reduction test [35]. The space between the two electrodes and the headspace available inside the Ziploc bag allowed for the reactive gas species to be generated. The modified atmospheric gas taken for the ACP treatment through DBD had higher percentages of oxygen, which leads to the generation of numerous free radicals with reactive oxygen species including ozone (O_3_), hydrogen peroxide (H_2_O_2_), atomic oxygen (O.), ozone radical ion (O_3−_), hydroperoxyl radical (HO_2_), superoxide anion (O_2−_), and others [34]. These gas species can react directly or indirectly with the molecular structure of the wheat flour and generate changes. Previous research has demonstrated that ACP treatment may change the molecular weights and solubility of wheat proteins. An increase in α-helix and β-turns and a decrease in β-sheets + antiparallel β-sheets in the secondary structure of the protein were determined in hard wheat flour and soft wheat flour with ACP treatment. A reduction in non-starch free fatty acids and phospholipids was further observed [31,32]. The rate of ozone generation was directly related to the rate of ACP treatment. The amount of ozone created for each sample was kept uniform by keeping a consistent volume in each package during ACP treatment.

### 4.2. Soft Wheat Flour Properties

Moisture content is an important parameter in HRC formulation to maintain consistency between flour batches. The moisture content of the different flour samples was in the range of 11.8–12.8% with no significant differences, as seen in Table 3. A study used heat treatment as an alternative to chlorine treatment and found that the moisture content of the flour needs to be adjusted for optimum baking results such as bulk density, cake height, texture, appearance, and mouth feel of the HRC [23,40]. However, the moisture content was not affected by ACP treatment; hence, no other adjustments were to be made. Similarly, there were no significant changes in the protein quantity of the flour at different oxidation treatments. The total protein content of soft wheat flour does not change due to oxidation, which aligns with previous research [16].

Hydrogen ions formed during the chlorination treatment (from Equation (1)) reduce the pH of wheat flour [19]. Chlorinated flour is nonselective and affects all types of flour, and soft wheat flour is typically treated to reach a target pH level of 4.4–4.8 [41]. In this study, the pH value of the chlorinated soft wheat flour was lowest while it was greatest for the untreated flour. The pH values of the ACP-treated flours showed a decreasing trend with increasing ACP voltage and time. With higher treatment conditions, the rate of oxidation is higher in the wheat flour, and the pH decreases. However, the pH at the highest rate of ACP treatment, i.e., 70 kV at 7 min, was still significantly greater than the chlorinated wheat flour sample. An increase in acidity, i.e., a decrease in the pH of wheat flour, helps in enhancing the flour quality for the baking of high-ratio cakes [41].

Hypochlorite ion formed during chlorination treatment is a strong oxidizing agent and acts on flour carotenoid pigments, which are part of lipid fraction. It reduces their levels rapidly, which bleaches the flour, thus increasing its whiteness [19]. The conjugated double bonds of the carotenoid molecules are affected by oxidation. Similar trends were observed in this study with ACP treatment of wheat flour, as shown in Table 4. Strong oxidation properties of ozone produced by ACP oxidize the double bonds in carotenoids. Previously, it was demonstrated that ACP treatment leads to an increase in the brightness of the color of the treated commodity [35]. The L* scale value ranged from 0 to 100, which was used to indicate the brightness or luminance; the a* value ranged from −60 to +60, indicating greenness to redness, and the b* value ranged from −60 to +60, indicating blueness to yellowness of the flour [42]. The differences in the color of the different treated samples can be determined with the ΔE value, which was calculated from the formula given in Equation (2). The ΔE values are evaluated as non-noticeable (0–0.5); slightly noticeable (0.5–1.5); noticeable (1.5–3); visible changes (3–6); and great changes (6–12) [37,40,43,44]. The changes in the ΔE values in different treatments are shown in Figure 4. 

From a one-way ANOVA test, there was a significant difference in the ΔE values in different treatments. The ΔE values shown in Table 4 show the different groups, which are significantly different from each other. The chlorinated wheat flour had the greatest change in color parameters as compared to the untreated samples. An increase in the brightness of the flours was observed with each increase in the rate of ACP application. Flours at the lowest rate of ACP treatment, i.e., 50 kV, had slightly noticeable changes; increasing ACP treatment to 60 kV and 70 kV resulted in noticeable changes. Previously, it was demonstrated an increase in brightness of whole wheat flour, which they treated with ACP at 80 kV for 30, 20, 10, and 5 min. The increase in the L* value with increasing treatment time is an indication of conjugated double bond degradation of yellow carotenoid pigments. This increase in flour brightness with ACP treatment was desirable as it enhances the brightness of the HRC being prepared [45].

### 4.3. Pasting Properties of Soft Wheat Flour

For wheat flour, when heated to a characteristic temperature in the presence of water, the native starch granules absorb the water and undergo swelling in a process called gelatinization. The gelatinization and swelling of wheat starch coincide with the loss of birefringence and crystallinity within the starch molecule. Swelling of the starch molecules results in the leaching of amylose, the linear fraction of the starch molecule in the aqueous suspension. The application of heat in the presence of excess liquid leads to the formation of viscous pastes, which are regarded as composite materials of the continuous polysaccharide phase. A sharp increase in the viscosity of the flour can be observed at a higher temperature called pasting temperature, and it characterizes the initiation of the pasting process [46].

A sucrose solution was used to determine the viscosity of the flour samples as it is more similar to an HRC batter environment. This is important for understanding how these samples will perform in an actual HRC system. High levels of sucrose, i.e., 50% (*w*/*w*), were used in our study, affecting starch swelling. The viscosity development of the untreated and ACP-treated occurred at the same temperature, but there were differences in initial viscosity, breakdown viscosity, and peak viscosity, which is shown in Table 5. Untreated wheat flour had the lowest viscosity development while chlorinated and ACP-treated wheat flour showed a higher viscosity. A higher water holding capacity was previously observed by Bosmans et al., 2019, in chlorinated flour as compared to untreated flour, which was correlated with prolonged viscosity rise during heating in the RVA [19].

There were significant differences in the peak viscosity of the different samples. The increase in peak viscosity of the wheat flours after ACP treatment might be due to the increase in crosslinking due to the disulphide bonds or changes in starch gelatinization. The water-holding capability of wheat flour increases due to the formation of disulphide bonds. The increase in viscosity also highlights changes in starch gelatinization. Thus, ACP treatment may induce changes in starch gelatinization properties of soft wheat flour [32]. Peak viscosity development in ACP-treated wheat flour showed an interesting pattern, where viscosity increased with each increase in time at a particular voltage. This was observed in the pasting values at 50 kV and 60 kV. However, the viscosity development at a greater voltage for shorter times was lower than the viscosity development at a lower voltage for a longer time. This indicates that the ACP treatment was time-dependent at a specific voltage. The effect of the time of treatment for each voltage is shown in Figure 5.

From Figure 5, it can be observed that peak viscosity development was greatest at 50 kV, 7 min followed by chlorinated flour. This might be due to more cross linking at higher rates of treatment, which might reduce its water-holding capability. The 50 kV, 7 min might have the optimum amount of cross linking developed in the flour to increase the viscosity [31,32]. Viscosity values for the ACP-treated soft wheat flours were greater than the untreated wheat flour in all cases. Thus, ACP generates changes in the starch pasting properties of flour. Viscosity development at a longer exposure to ACP at 50 kV, 60 kV and 70 kV was similar to that observed for increasing chlorination treatment. ACP treatment appears to elicit similar changes in the pasting behavior of soft wheat flour, such as chlorination.

### 4.4. High-Ratio Cake Properties

Cake volume is an indicator of the size and overall quality of the HRCs. Volume can be affected by flour quality, batter viscosity, stability, and specific gravity during mixing and baking. The amount of air and CO_2_ captured during mixing, as well as the amount of air, moisture, and carbon dioxide entrapped and expanded during baking, affects volume. The volume of an HRC is not always indicative of its desirability; however, a lower volume indicates a heavier and less desirable crumb [47]. Chlorinated flour is typically used in the preparation of HRCs as it enhances the ability of the flour to carry more water, sugar, and fat than untreated wheat flour. HRCs with much larger volume and softer texture have been produced using wheat flour treated with ozone [48]. The volume of the HRC in this study was also positively impacted by ACP treatment.

HRCs prepared with chlorinated flour had a greater volume than those made with untreated flour, and it was also noted that the volume of HRCs increased with ACP treatment. The volume of the HRC was highest at 60 kV, 6 min and 70 kV, 6 min. This might be due to the trend of greater water retention capacity with increasing levels of oxidation. The volume of the HRC at 60 kV, 7 min and 70 kV, 7 min was observed to be lower as compared to the same treatments at 5 min and 6 min. This might be due to excessive oxidation of protein molecules at higher rates of treatment, which might have implications for its water retention [31]. The HRC from untreated soft wheat flour had a very low rise as compared to chlorinated and other ACP-treated soft wheat flours.

Figure 6 shows different HRCs baked with untreated flour, chlorinated flour, and different ACP treatments. It can be observed that the appearance of the HRCs was different under different treatment conditions.

A uniform texture with fine grain properties is desirable in an HRC. When the batter is heated, it undergoes two significant physicochemical changes: starch gelatinization and protein denaturation. However, once the HRC has been removed from the oven and allowed to cool, the gases condense, and the structural strength of the crumb is determined. If the crumb structure is weak, the gas cells compress, resulting in an unevenly textured crumb [16]. When all of the gas is compressed, waxy cores can form in the upper crust’s center. As a result, texture is an important HRC quality parameter that is carefully monitored. Chlorination prevents the collapsing of HRCs and helps in maintaining a fine texture [16]. In Figure 3, the texture parameters defined by the firmness of the HRCs with untreated wheat flour, chlorinated flour, and different ACP treatments are shown. The firmness of HRCs with untreated wheat flour was higher compared to chlorinated wheat flour and ACP-treated wheat flour. There was a significant statistical difference in the firmness of HRCs prepared from untreated flour, chlorinated flour, and ACP-treated flour. Chlorinated wheat flour HRCs had an average firmness of 577.33 g while the lowest was observed for ACP-treated wheat flour at 60 kV, 5 min of treatment with an average of 447.09 g firmness. However, the firmness of the HRCs increased at higher treatment rates from 60 kV, 5 min. At 70 kV, 7 min, the firmness of the HRCs was at an average of 669.99 g, which was even higher than that of the untreated wheat flour HRC. With the increase in treatment rates, the firmness of the HRC increased. This aligns with the other tests that showed that higher rates of ACP treatment were undesirable for baking HRCs.

### 4.5. C-Cell Structure of the High-Ratio Cakes

Cell structure imaging of the crust and crumb was performed to understand the impact of the treatments on the cell parameters, such as the number of cells, area of cells in percentage, number of holes, area of holes in percentage, volume of cells, non-uniformity of the cells, cell diameter, and crumb fineness on a defined area of the HRCs. Cell structure imaging also helps in determining the brightness of the slice and fineness of the crumb along with other factors. This information helps in correlating the volume and scoring of the HRC, which was usually carried out manually. It can be seen in Table 6 that the average cell diameter was lowest for HRCs prepared from chlorinated soft wheat flour and highest for untreated soft wheat flour. The 50 kV, 6 min had the least cell diameter compared to the other rates of ACP treatment. The average cell diameter of chlorinated flour was 3.23 mm; for untreated flour, it was 5.41 mm. Increased treatment rates led to a more porous structure of the crumb in HRCs.

## 5. Conclusions

This study confirms that ACP generates changes in soft wheat flour properties and HRCs prepared from treated flour. A significant reduction in pH values was observed with ACP treatment due to the strong oxidation potential of ACP. Peak viscosity measurements were highest at 50 kV, 7 min, which highlights changes in starch gelatinization in wheat flour due to ACP treatment. These changes were also reflected in HRCs prepared from ACP-treated flour. The highest volume of the HRCs was observed at 70 kV, 6 min, and the lowest firmness was recorded at 60 kV, 7 min. C-cell analysis indicated that ACP treatment can improve the uniformity of cells in the cake crumb. Low rates of ACP treatment such as 50 kV, 6 min and 50 kV, 7 min may be used for the treatment of soft wheat flour. High rates of treatment may generate undesirable cake texture and flavor properties due to excessive oxidation. ACP has great potential as an alternative to chemical oxidative agents used in wheat flour processing. Further optimization of ACP parameters could potentially match or surpass the favorable qualities from chlorinated flour and have wider applications in the food industry.

## Figures and Tables

**Figure 1 foods-13-02366-f001:**
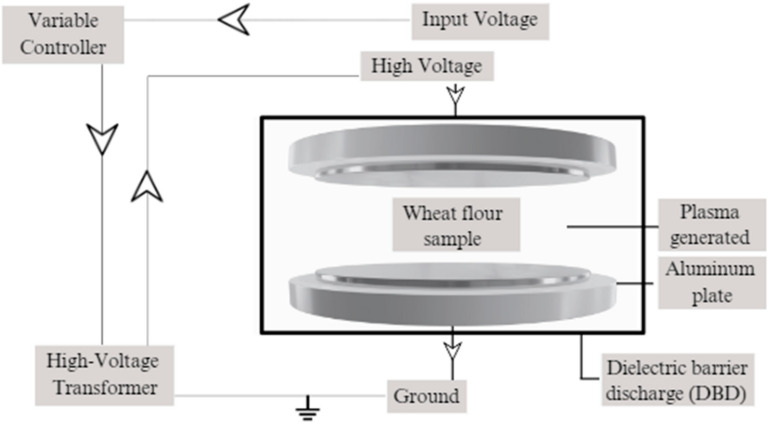
Schematic diagram of atmospheric cold plasma system to treat wheat flour. Note that the image is not to scale.

**Figure 2 foods-13-02366-f002:**
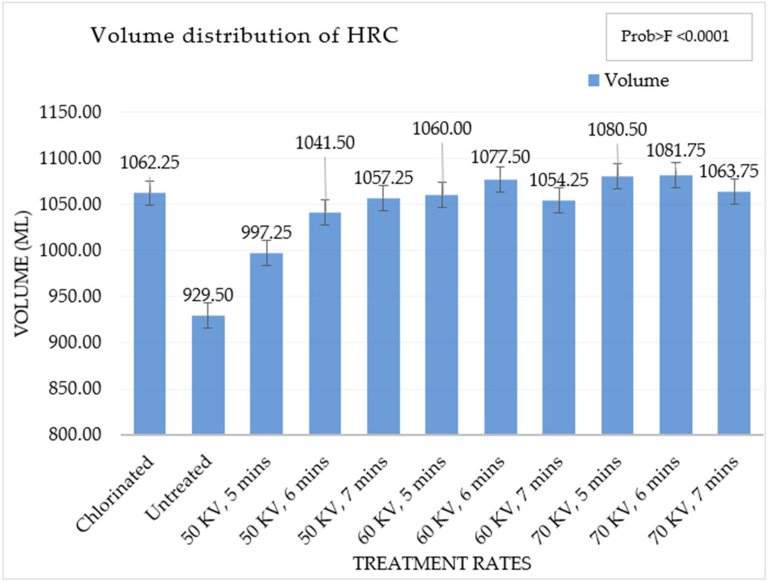
Volume of high-ratio cakes (mL) in different treatments.

**Figure 3 foods-13-02366-f003:**
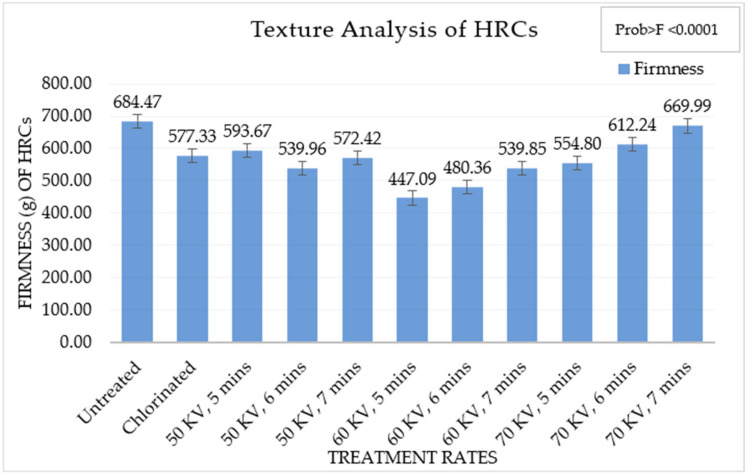
Firmness of high-ratio cakes (g) in different treatments.

**Figure 4 foods-13-02366-f004:**
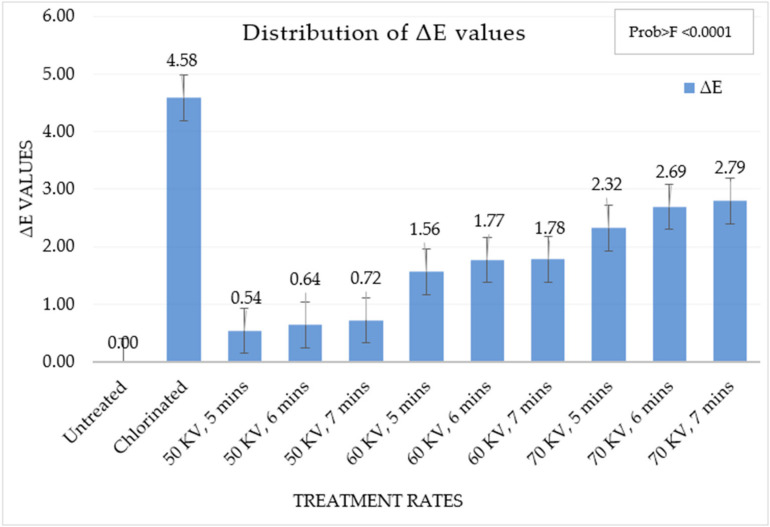
ΔE values distribution in different treatments.

**Figure 5 foods-13-02366-f005:**
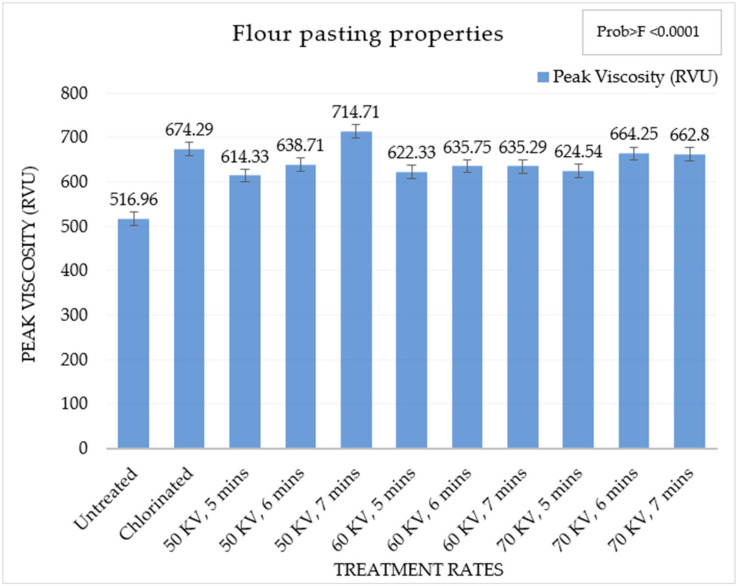
Pasting behavior of soft wheat flour samples at different treatments.

**Figure 6 foods-13-02366-f006:**
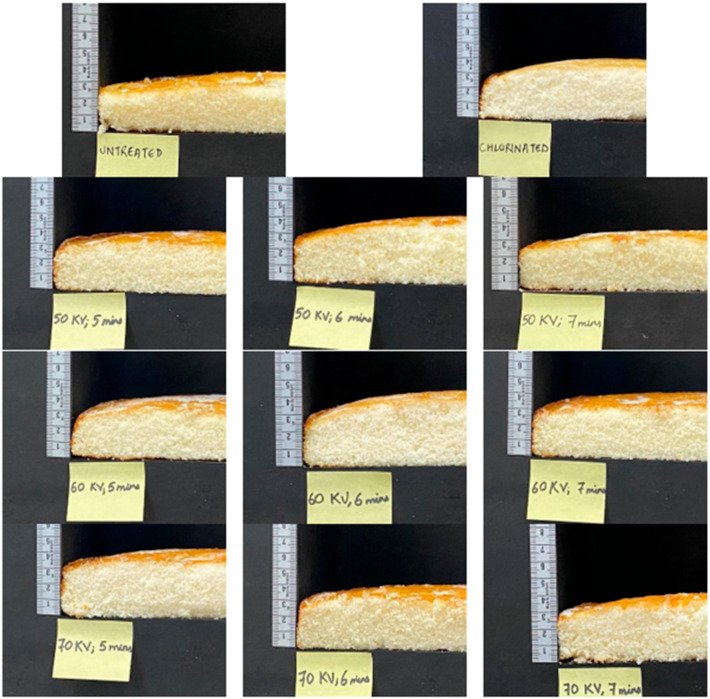
High-ratio cakes under different treatments.

**Table 1 foods-13-02366-t001:** RVA internal bake temperature profile.

Elapsed Time	Temperature	Speed
0:00	25 °C	960 rpm
0.10	-	160 rpm
1:00	40 °C	160 rpm
2:00	50 °C	160 rpm
3:00	60 °C	160 rpm
4:00	70 °C	160 rpm
5:00	78 °C	160 rpm
6:00	84 °C	160 rpm
7:00	88 °C	160 rpm
8:00	92 °C	160 rpm
9:00	95 °C	160 rpm
10:00	98 °C	160 rpm
11:00	99 °C	160 rpm
20:00	99 °C	160 rpm

**Table 2 foods-13-02366-t002:** Formulation of high-ratio cake.

Ingredients	% (Flour Basis)
Flour	100.00
Sugar	140.00
Shortening	50.00
Nonfat dry milk	12.00
Dried egg whites	9.00
NaCl	3.00
Baking powder	6.50
Water	125.00

**Table 3 foods-13-02366-t003:** Moisture, protein, and pH contents of soft wheat flour samples.

Treatment	Moisture (%)	Protein (%)	pH
Untreated	11.91 _a_	9.29 _a_	6.14 _b_
Chlorinated	11.49 _a_	9.27 _a_	4.44 _b,c,d,e_
50 kV, 5 min	11.97 _a_	9.36 _a_	6.08 _b,c_
50 kV, 6 min	12.13 _a_	9.38 _a_	6.04 _b,c_
50 kV, 7 min	12.27 _a_	9.30 _a_	6.03 _b,c_
60 kV, 5 min	12.22 _a_	9.37 _a_	5.99 _c_
60 kV, 6 min	12.19 _a_	9.33 _a_	6.04 _b,c_
60 kV, 7 min	12.33 _a_	9.37 _a_	5.79 _d_
70 kV, 5 min	12.34 _a_	9.33 _a_	5.72 _d_
70 kV, 6 min	12.48 _a_	9.37 _a_	5.58 _e_
70 kV, 7 min	12.52 _a_	9.35 _a_	5.43 _e_

Note—Within each column, values with the same letter (_a,b,c,d,e_) show no significant difference, whereas values with different letters show significant difference. Moisture: *p* = 0.4122, no significant difference; protein: *p* = 0.2072, no significant difference; and pH: *p* < 0.001, significant difference.

**Table 4 foods-13-02366-t004:** *L*a*b* measurements of soft wheat flour samples.

Treatment	L	a	b	ΔE
Untreated	92.16 _i_	−1.93 _i_	7.22 _b_	0.00 _a_
Chlorinated	92.27 _i_	−0.76 _i_	2.79 _a_	4.58 _b_
50 kV, 5 min	92.11 _i_	−1.82 _i_	6.69 _b_	0.54 _c_
50 kV, 6 min	92.19 _i_	−1.80 _i_	6.59 _b_	0.64 _c_
50 kV, 7 min	92.20 _i_	−1.77 _i_	6.52 _b_	0.72 _c,d_
60 kV, 5 min	92.30 _i_	−1.59 _i_	5.70 _c_	1.56 _e_
60 kV, 6 min	92.19 _i_	−1.51 _i_	5.50 _c_	1.77 _f_
60 kV, 7 min	92.17 _i_	−1.51 _i_	5.49 _c_	1.78 _f_
70 kV, 5 min	92.27 _i_	−1.42 _i_	4.95 _c_	2.32 _g_
70 kV, 6 min	92.35 _i_	−1.33 _i_	4.60 _c_	2.69 _h_
70 kV, 7 min	92.38 _i_	−1.32 _i_	4.50 _c_	2.79 _h_

Note—Within each column, _a,b,c,d,e,f,g,h_: *p* < 0.0001, significant difference, _i_: no significant difference. Means indicated by the same alphabet are not significantly different.

**Table 5 foods-13-02366-t005:** Pasting properties of soft wheat flour samples.

Treatment	Initial Viscosity (RVU)	Trough 1 (RVU)	Breakdown Viscosity (RVU)	Peak Viscosity (RVU)
Untreated	4.17 _a_	205.00 _c_	305.25 _b_	516.96 _a_
Chlorinated	8.58 _b_	357.00 _a_	300.25 _b_	674.29 _b,c_
50 kV, 5 min	7.92 _b_	263.33 _a_	345.42 _a,b_	614.33 _c_
50 kV, 6 min	7.83 _b_	277.42 _a_	355.00 _a_	638.71 _b,c_
50 kV, 7 min	12.25 _c_	417.75 _b_	299.50 _b_	714.71 _d_
60 kV, 5 min	7.83 _b_	279.50 _a_	327.83 _b_	622.33 _b_
60 kV, 6 min	7.08 _b_	335.75 _a_	321.25 _b_	635.75 _b,c_
60 kV, 7 min	8.00 _b_	261.83 _a_	279.92 _b_	635.29 _b,c_
70 kV, 5 min	4.08 _a_	277.92 _a_	330.58 _b_	624.54 _b,c_
70 kV, 6 min	7.83 _b_	326.83 _a_	358.50 _a_	664.25 _b,c_
70 kV, 7 min	8.08 _b_	315.50 _a_	333.83 _b_	662.80 _b,c_

Note—Within each column, _a,b,c,d_: *p* < 0.0001, significant difference. Means indicated by the same alphabet are not significantly different.

**Table 6 foods-13-02366-t006:** C-cell analysis of high-ratio cakes at different treatments.

Treatment	Cell Diameter (mm)
CHLORINATED	3.23 _a_
UNTREATED	5.41 _c_
50 kV, 5 min	4.59 _b,c_
50 kV, 6 min	4.08 _a,b_
50 kV, 7 min	4.49 _b,c_
60 kV, 5 min	4.19 _a,b,c_
60 kV, 6 min	4.81 _b,c_
60 kV, 7 min	4.59 _b,c_
70 kV, 5 min	4.95 _b,c_
70 kV, 6 min	5.17 _b,c_
70 kV, 7 min	4.95 _b,c_

Note—_a,b,c_: *p* = 0.0018; i.e., significant difference. Means indicated by the same alphabet are not significantly different.

## Data Availability

Data supporting the reported results can be found in the Texas A&M University OakTrust Data Repository (Mahanta et al., 2024).

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
