# Peer review of "Atmospheric Cold Plasma as an Alternative to Chlorination in Soft Wheat Flour to Prepare High-Ratio Cakes"

_foods, 2024, doi:10.3390/foods13152366_

Round 1

Reviewer 1 Report

Comments and Suggestions for Authors

1. The standard deviation should be inserted in all tables and the results in Table 3 should be analyzed.

2. The results should be put together with the discussion, and the grammar of the article needs to be improved.

3. A clear and in-depth analysis of the possible ways in which ACP alter bread quality should be provided.

Comments on the Quality of English Language

The quality of English language should be modified.

Author Response

Reviewer 1 Comments

The standard deviation should be inserted in all tables and the results in Table 3 should be analyzed.

Statistical significance has been added in the tables and table 3 is updated.

The results should be put together with the discussion, and the grammar of the article needs to be improved.

The format of the paper requires separate results and discussion

A clear and in-depth analysis of the possible ways in which ACP alter bread quality should be provided.

The discussions and conclusion is updated with a more in depth effect of ACP on high ratio cake quality.

Reviewer 2 Report

Comments and Suggestions for Authors

As an emerging technology, the potential of atmospheric cold plasma as the alternative to chlorination for high ratio cakes was researched in this work. The manuscript is interesting and novel, it is also scientific sound and could be interesting for readers of journal. This work needs a moderate revision. And the detailed comments are listed as follows.

1. Line 157, for the modified atmospheric gas, where were the mixed gases get from? Why to choose it? What is volume of the gases filled in LDPE plastic bag? Air might be employed as an alternative for reaction gas.

2. Line 166, the figure title should be moved below.

3. Line 389, only one trained panelist? The evaluation must be representative for result statistic.

4. Table 7, the flavor scores of the ACP treatments were 0 excluding 50 kV for 5 min, indicating that the ACP treatment might be not suitable for real-world cake production. Flavor is very important. Please explain it.

5. In ACP treatment of soft wheat flour section of Discussion, please explain in detail the mechanism of wheat flour molecular structure change by ACP.

6. In Table 3, the pH of 60 kV for 6 min were observably higher than the pH of 60 kV for 7 min, it was different from the interpretation in Line 440-441. Please explain it.

7. The line 538-540, the explanation “it can be observed that peak viscosity development was greatest at 50 kV, 7 minutes and 60 kV, 7 minutes, even greater than that for chlorination.” Actually, the peak viscosity at 60 kV, 7 minutes was lower than 70 kV, 6 or7 minutes. Please check the result.

8. For Figure 6, please provide more clear pictures for identification.

9. What is the ACP treatment condition for proposal? Why?

10. The conclusion is too short for summary, and some important results should be supplemented and discussed for the prospect. And the shortcomings must be also discussed for ACP in this manuscript.

Comments on the Quality of English Language

Quality of English Language is fine.

Author Response

Reviewer 2 Comments

Line 157, for the modified atmospheric gas, where were the mixed gases get from? Why to choose it? What is volume of the gases filled in LDPE plastic bag? Air might be employed as an alternative for reaction gas.

The modified atmospheric gasses were obtained from Airgas, Inc. Previous research has demonstrated that we want higher concentration of oxygen, which gets oxidized to ozone. The volume inside the LDPE bag was 26.6X27.3X3.5 cm3. The gas combination may be chosen on the basis of desired ionized gas, here ozone was used for the treatment and analysis. It has been updated in the paper.

Line 166, the figure title should be moved below.

It has been formatted

Line 389, only one trained panelist? The evaluation must be representative for result statistic.

There were 3 trained panelists, it has been updated.

Table 7, the flavor scores of the ACP treatments were 0 excluding 50 kV for 5 min, indicating that the ACP treatment might be not suitable for real-world cake production. Flavor is very important. Please explain it.

An explanation was added to address the potential of masking the flavor after ACP treatment in the conclusion. The sensorial evaluation tests and results has been removed.

In ACP treatment of soft wheat flour section of Discussion, please explain in detail the mechanism of wheat flour molecular structure change by ACP.

The changes in molecular structures of different components has been explained in the paper.

In Table 3, the pH of “60 kV for 6 min” were observably higher than the pH of “60 kV for 7 min”, it was different from the interpretation in Line 440-441. Please explain it.

It is mentioned that with higher treatment, the rate of oxidation is higher and the pH decreases. Since 60 KV, 6 minutes is a lower treatment rate than 60 KV, 7 it has a higher pH than 60 Kv, 7 minutes

The line 538-540, the explanation “it can be observed that peak viscosity development was greatest at 50 kV, 7 minutes and 60 kV, 7 minutes, even greater than that for chlorination.” Actually, the peak viscosity at 60 kV, 7 minutes was lower than 70 kV, 6 or7 minutes. Please check the result.

The results are changed to highest viscosity at 50kV, 7 minutes and followed by chlorination.

For Figure 6, please provide more clear pictures for identification.

The pictures are formatted better but they are the same as before.

What is the ACP treatment condition for proposal? Why?

Low rates of ACP treatment such as 50 kV, 6 minutes and 50 kV, 7 minutes may be used for treatment of soft wheat flour as it positively affects different parameters and the undesirable flavor may be masked using a certain flavor.

The conclusion is too short for summary, and some important results should be supplemented and discussed for the prospect. And the shortcomings must be also discussed for ACP in this manuscript.

The conclusion is updated with detailed  explanation.

Reviewer 3 Report

Comments and Suggestions for Authors

Dear authors, after reviewing the article submitted with id foods-3090667 I consider it important to mention the following points with the idea of contributing to the improvement of your proposal:

Line 70, "rich" in what sense?

Line 86, the citation mode is correct?

Line 111, after reading the reference article for the quotation I consider that there are other more recent articles that are more in line with your topic, please consider it.

Line 119, "no hazardous residues" requires a citation so that it is not an assumption.

Line 125, missing a period in the citation.

Lines 159 and 160, it is recommended to give more details of the reasons why you selected those variables, it has to be clear.

Line 186, where was the sample obtained from? please indicate it.

Line 213, is Perten

Line 233, sea in lowercase letters

Line 235, keep the same format in all your tables, figures and images.

Line 304, table 4, why the color coordinates do not have the statistics applied? also the averages do not show the error or standard deviation.

Remark: From the results onwards the text is no longer justified, please present a homogeneous format.

Line 323, I recommend presenting the information in a graph so that the differences can be easily and quickly observed by the reader.

Line 327, the samples were shaken at 160 rpm that really mimics the process? I think it is not the correct term, I would recommend to rethink the idea.

Line 343, please tables and graphs should have the same order of presentation of the samples so as not to lose the reader.

Line 401 onwards, the discussion of the results is minimal, there are sections where they again explain the fundamentals of a phenomenon but do not explain it, for example in line 626 it is for discussion, not for the technique. I strongly recommend the authors to rewrite the discussion of results section and avoid just repeating the data obtained and say if they are statistically significant or not, they should also give the probable causal factors, change some references to recent ones otherwise the impact of their work is lost. 

The positive part is that their proposed work opens the way to ask what happens with the macromolecules within the system and what happens at the nutritional level if a product like this is consumed. 

Author Response

Reviewer 3 Comments

Line 70, "rich" in what sense?

The statement has been updated and the batter characteristics are refined as “smooth”.

Line 86, the citation mode is correct?

It is resolved.

Line 111, after reading the reference article for the quotation I consider that there are other more recent articles that are more in line with your topic, please consider it.

A recent publication has been cited.

Line 119, "no hazardous residues" requires a citation so that it is not an assumption.

A citation has been added.

Line 125, missing a period in the citation.

It has been added.

Lines 159 and 160, it is recommended to give more details of the reasons why you selected those variables, it has to be clear

It has been mentioned why the rates of ACP treatment was used for the wheat flour.

Line 186, where was the sample obtained from? please indicate it.

The source of wheat flour is mentioned in line 142-145.

Line 213, is Perten

It has been corrected.

Line 233, sea in lowercase letters

It has been corrected.

Line 235, keep the same format in all your tables, figures and images.

The tables, figures and images are formatted.

Line 304, table 4, why the color coordinates do not have the statistics applied? also the averages do not show the error or standard deviation.

The table has been updated with statistics.

Remark: From the results onwards the text is no longer justified, please present a homogeneous format.

Texts are formatted.

Line 323, I recommend presenting the information in a graph so that the differences can be easily and quickly observed by the reader.

The peak viscosity is shown in graph in figure 5.

Line 327, the samples were shaken at 160 rpm that really mimics the process? I think it is not the correct term, I would recommend to rethink the idea.

When we say we are mimicking the process with the RVA, we mean that we are mimicking the internal temperature profile of a cake during baking. The rotor must spin at 160 rpm in order to detect the changes in viscosity during heating. Of course, the use of a rotor spinning at 160 rpm wouldn’t be part of the process in a cake during baking – it would destroy the structure. However, it is a necessary part of this test to shear the sample under temperature/time conditions that are similar to those for HRCs during baking so that we can gain insight into the potential changes in viscosity (and hence starch gelatinization characteristics) induced by ACP treatment. These changes ultimately influence HRC volume and texture.

Line 343, please tables and graphs should have the same order of presentation of the samples so as not to lose the reader.

The tables, figures and images are formatted.

Line 401 onwards, the discussion of the results is minimal, there are sections where they again explain the fundamentals of a phenomenon but do not explain it, for example in line 626 it is for discussion, not for the technique. I strongly recommend the authors to rewrite the discussion of results section and avoid just repeating the data obtained and say if they are statistically significant or not, they should also give the probable causal factors, change some references to recent ones otherwise the impact of their work is lost.

The discussion is updated and the p values are mostly removed from the discussion. 

Round 2

Reviewer 1 Report

Comments and Suggestions for Authors

Authors have modified the paper carefully, I recommend the acceptance of the paper at present status.